

# Spatial Relationship between Precipitation and Runoff in Africa

Fidele Karamage[1,2,3,4], Yuanbo Liu[1], Xingwang Fan[1], Meta Francis Justine[2,5], Guiping Wu[1], Yongwei Liu[1], Han Zhou[1], Ruonan Wang[1]

[1]Key Laboratory of Watershed Geographic Sciences, Nanjing Institute of Geography & Limnology, Chinese Academy of
Sciences, Nanjing 210008, People's Republic of China
[2]University of Chinese Academy of Sciences, Beijing 100049, China
[3]University of Lay Adventists of Kigali, P.O. Box 6392, Kigali, Rwanda
[4]Joint Research Center for Natural Resources and Environment in East Africa, P.O. Box 6392, Kigali, Rwanda
[5]Chengdu Institute of Biology, Chinese Academy of Sciences, Chengdu 610041, People's Republic of China

*Correspondence to*: Yuanbo Liu (ybliu@niglas.ac.cn and yb218@yahoo.com)

**Abstract.** Lack of sufficient and reliable hydrological information is a key hindrance to water resource planning and management in Africa. Hence, the objective of this research is to examine the relationship between precipitation and runoff at three spatial scales, including the whole continent, 25 major basins and 55 countries. For this purpose, the long-term monthly runoff coefficient ($R_c$) was estimated using the long-term monthly runoff data (R) calculated from the Global Runoff Data
Centre (GRDC) streamflow records and Global Precipitation Climatology Centre (GPCC) precipitation datasets for the period of time spanning from 1901 to 2017. Subsequently, the observed $R_c$ data were interpolated in order to estimate $R_c$ over the ungauged basins under guidance of key runoff controlling factors, including the land-surface temperature (T), precipitation (P) and potential runoff coefficient ($C_o$) inferred from the land use and land cover, slope and soil texture information. The results show that 16% of the annual mean precipitation (672.52 mm) becomes runoff (105.72 mm), with a runoff coefficient of 0.16,
and the remaining 84% (566.80 mm) evapotranspirates over the continent during 1901 – 2017. Spatial analysis reveals that the precipitation-runoff relationship varies significantly among different basins and countries, mainly dependent on climatic conditions and its inter-annual variability. Generally, high runoff depths and runoff coefficients are observed over humid tropical basins and countries with high precipitation intensity compared to those located in subtropical and temperate drylands.

**Keywords**: Africa; basin; evapotranspiration; GIS; precipitation; runoff coefficient; water balance.

**1 Introduction**

In the 21st century water resources management becomes a major concern to human life and environmental protection (Cosgrove and Loucks, 2015). It is well known that precipitation is the source of freshwater on our planet, and its intensity varies from one region to another. Lacking precipitation often causes droughts which would further induces severe environment degradation, social conflicts and hunger crisis (Messer et al., 2001; Clover, 2003). Precipitation-to-runoff is the
main source of water for rivers, lakes and ocean replenishment (Edwards et al., 2015). Precipitation scarcity aggravates poverty



to an estimate of 300 million people living in the Eastern and Western drylands of Africa and the number is expected to increase by 65-80% in 2030 (Cervigni and Morris, 2016). By 2050, it is estimated that 40% of the global population will be exposed to river basins that experience severe water stress, particularly in Africa and Asia (UNISDR, 2015). On the other hand, storm water-runoffs cause significant hazards and disasters such as soil erosion, floods, landslides, water pollution, and

infrastructure damage (Goudie, 2000; Weng, 2001; Karamage et al., 2017a). For instance, the population exposed to flood threats increased from 0.5 to 1.8 million between 1970 and 2010 in Sub-Saharan Africa (UNISDR, 2011). Water-related problems have far-reaching effects in Africa, where, limited financial funds, sparse hydrological data and reliable scientific information bothers sustainable planning and management of water resources and related disasters (Oyebande, 2001; Karamage et al., 2016; Urroz et al., 2001).

Although runoff studies have been conducted at global scale and in some local areas in Africa (Hong et al., 2007; Fekete et al., 2002), there is no study yet indicating the spatial relationship between precipitation and runoff within all African basins and countries. In this context, current study analyzed the precipitation-runoff relationship, using an indicative runoff coefficient within 25 major basins and 55 countries in Africa. Besides, this study proposed a novel method for estimating runoff coefficient over ungauged areas based on the environmental characteristics of gauged and ungauged basins. This method

derived runoff coefficient by taking consideration of its major controlling factors, which might be useful to the scientists and water resource planners. The runoff coefficient is the ratio of runoff depth to rainfall intensity within a specific watershed (Kadioglu and ŞEN, 2001) and it varies between 0 and 1 (Blume et al., 2007). It is used to indicate how much waterflow converted from precipitation within a given time and catchment. In addition, runoff coefficient is very useful for catchment scale land use and flood management in any catchment (Sriwongsitanon and Taesombat, 2011). Geographical Information

System (GIS) has evolved since its introduction in the 1960s, and now becomes a widely used tool able to deal with multiple variables regarding basin management. However, hydrological GIS-based studies rely strongly on databases (Terakawa, 2003). In this respect, this study generated $R_c$ based on monthly runoff calculated from the Global Runoff Data Centre (GRDC) discharges (GRDC, 2018) and Global Precipitation Climatology Centre (GPCC) precipitation products for 1901 – 2017 (Becker et al., 2013; Schamm et al., 2014) in gauged basins. The $R_c$ data were then interpolated to the ungauged areas using

the key factors such as land-surface temperature (T), Precipitation (P) and potential runoff coefficient ($C_o$) estimated from the land use and land cover, soil texture and slope information by using GIS spatial analysis techniques. These environmental factors are critical to the estimation of runoff coefficient (Sriwongsitanon and Taesombat, 2011; Chen et al., 2007) . Impervious surfaces generally correspond to higher runoff coefficient and larger runoff volume than previous surfaces (Weng, 2001). Urbanization and cropland expansion reduces infiltration capacity and boosts the generation of surface water runoff (Goudie,

2000; Weng, 2001). Evapotranspiration is generally less than precipitation in wet seasons, that is positive water balance due to groundwater accumulation, which results in an increased surface runoff. In dry seasons evapotranspiration exceeds precipitation because the plants absorb underground water and cause a water deficit. However, underground water can be ignored in the long-term annual mean water balance estimation due to a variety of wet and dry seasons (Long et al., 2014).



## 2 Data inputs and Methods

### 2.1 Study area

Africa (Figure 1) is the world's second-largest continent (≃ 30.3 million km²) accounting for 6% of Earth's surface area and 20.4 % of land area (Sayre and Pulley, 1999; Mawere, 2017). It is the second-most-populous continent (1,256 million

5    people) after Asia (4,504 million people) as of 2017 (UN-DESA, 2017).



**Figure 1.** Hydrological map showing major rivers, lakes, 25 major basins and 55 countries of Africa.



The European Space Agency (ESA) Climate Change Initiative (CCI) land cover (LC) map 2015 (ESA-CCI, 2017) indicates that Africa is comprised of forests (24.52%), grassland (24.51%), cropland (16.14%), built-up areas (0.16%), wetlands (0.84%), inland water (0.99%), and bare areas (32.84%). Over 60% of the soil is dominated by hot, arid or immature soil assemblages: Arenosols (22%), Leptosols (18%), Cambisols (11%), Calcisols (5%), Regosols (3%) and Solonchacks/Solonetz (2%). Another 20% is characterized by tropical or sub-tropical features: Ferralsols (10%), Plinthisols (5%), Lixisols (4%) and Nitisols (2%) (Dewitte et al., 2013). Based on the Climatic Research Unit Timeseries (CRU TS) land-surface temperature dataset (1901 – 2016) (Harris et al., 2014), and the GPCC dataset (1901 – 2017) (Becker et al., 2013; Schamm et al., 2014), the African continent has an overall mean annual surface temperature (T) of 24°C and mean precipitation (P) of 672.52 mm. It has three major climate types including tropical (T = 25°C; P = 835 mm·yr$^{-1}$), subtropical (T = 22°C; P = 156 mm·yr$^{-1}$) and temperate zones (T = 18°C; P = 261 mm·yr$^{-1}$). The topography is characterized by large-scale extensional features such as the East African Rift, anomalously subsided basins and uplifted domes (Moucha and Forte, 2011). The continent is divided into 25 major hydrological basins according to its hydrological characteristics (FAO, 2009). Drainage patterns are controlled by the distribution of basins and swells, about 95% is drained through permanent or ephemeral rivers. However, in arid areas (i.e., Northwest Sahara Desert and Somalia), drainage is sometimes absent or masked by sand seas. Approximately, 60% of the African continent is drained by 10 large rivers (Congo, Limpopo, Niger, Nile, Ogooue, Orange, Senegal, Shebelle, Volta and Zambezi) and their tributaries (Paul et al., 2014).

## 2.2 Datasets and Application

As presented by a conceptual framework (Figure 2), the goal of this study is achieved primarily using two types of data inputs (river discharge data and precipitation) in monitored basins and further auxiliary datasets comprising the land-surface temperature, precipitation and potential runoff coefficient established based on the land use and land cover (LULC) maps, Digital Elevation Model (DEM) and soil properties employed to improve the interpolation accuracy of observed runoff coefficient ($R_c$). The data are processed and analyzed using the Esri ArcGIS software version 10.5, SDMToolbox version 2.2 (Brown et al., 2017) and Excel VBA (Visual Basic for Applications) (Walkenbach, 2010).



**Figure 2.** A conceptual framework used for analyzing the precipitation-runoff relationship in Africa.

### 2.2.1 Runoff coefficient estimation in gauged basins

The runoff coefficient in monitored basins is estimated from two types of data: (1) the monthly time series of river discharge data for 341 African river basins (1901 – 2017) provided by request from the Global Runoff Data Centre (GRDC) (GRDC, 2018) and (2) monthly precipitation datasets acquired from the Global Precipitation Climatology Project (GPCP) (Becker et al., 2013). The GRDC  is an international organization based in Germany, a branch of the World Meteorological





Organization (WMO) that was established in 1988 to support scientific studies on global climate change and water resources management (GRDC, 2018). Figure 3 shows ungauged areas and gauged catchments of Africa under consideration of a complete set of 12 months in a year (from January to December) during the period of 117 years (1901 – 2017). The shapefile of 25 major African basins was provided by the Food and Agriculture Organization of the United Nations (FAO)  (FAO, 2009).

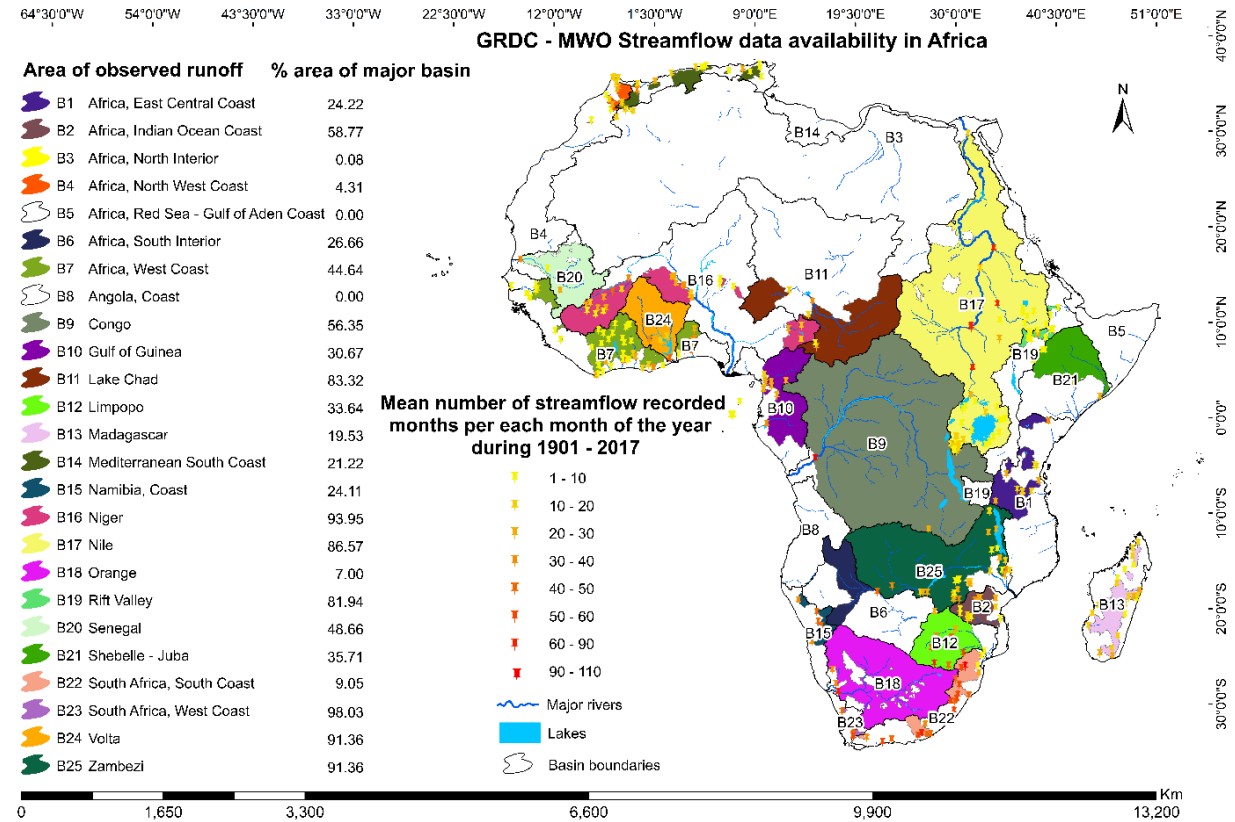

**Figure 3.** Land area coverage of observed GRDC-WMO streamflow records within 25 major basins of Africa.

On account of discharge data that were discontinuously recorded in different months and various stations, and which could not allow the possibility of monthly or annual trend analysis, the present study was carried out using the long-term mean monthly runoff and their corresponding rainfall input for the period 1901 – 2017. This method enabled us to obtain the streamflow records for all 12 months of the year and maximize the number of monitored basins, covering a total surface area of 12.37 million km$^2$ ($\simeq$ 41.4% of the total African continent). Given that, streamflow records were provided in cubic meter per second (m$^3$·S$^{-1}$), monthly runoff depths at basin scale were calculated by Eq. (1):

$$R = \frac{(Q \cdot 1000 \cdot 24 \cdot n \cdot 3600)}{A} \tag{1}$$

where, R is an average runoff depth (mm·month$^{-1}$) for the drainage area or basin of interest A; A is a drainage area (basin) in m$^2$; Q is river discharge (m$^3$·S$^{-1}$) drained from the basin of interest A (m$^2$); n is the number of days in each month.



The long-term mean precipitation was computed from the Global Precipitation Climatology Centre (GPCC) version 7.0 at 0.5° resolution (Becker et al., 2013). This is the centennial GPCC Full Data Reanalysis of monthly global land-surface precipitation with a duration record of 10 years or longer from 75,000 stations worldwide. The temporal coverage of the dataset ranges from January 1901 to December 2013. The GPCC Full Data Reanalysis is the most accurate in situ precipitation dataset which

supports studies on regional climate monitoring, model validation, and water resource assessment (Becker et al., 2013). The remaining period starting from 2014 to 2017 was completed by monthly total precipitation calculated and resampled at 0.5° from the GPCC First Guess Product at 1° resolution of daily global land-surface precipitation based on the station database (SYNOP) available via the Global Telecommunication System (GTS) of the World Meteorological Organization (WMO) at the time of analysis (3 – 5 days after the end of the analysis month). This product contains the daily totals for a month on a

regular latitude/longitude grid with a spatial resolution of 1° x 1° latitude by longitude (Schamm et al., 2014).

By following Eq. (2)  (Kadioglu and ŞEN, 2001), we estimated the long-term monthly runoff coefficient as the ratio of long-term runoff depth to long-term monthly precipitation for the study period 1901 – 2017.

$$R_c = \frac{R}{P} \tag{2}$$

where, $R_c$ is a runoff coefficient (dimensionless); R is a runoff depth (mm) and P is precipitation intensity (mm).

**2.2.2 Runoff coefficient estimation in ungauged basins**

Numerous studies have established different climatic geospatial datasets such as precipitation and surface temperature, etc., using interpolation algorithms based on a certain number of recorded locations directed by the other auxiliary variables (i.e.: DEM) to improve the results (Ahmed et al., 2014; Huang and Hu, 2009; Sanabria et al., 2013). In this sense, the present study developed a hybrid interpolation method to estimate the $R_c$ in ungauged basins of Africa using the Inverse

Distance Weighting (IDW) interpolation algorithm directed by major runoff controlling factors (Potential runoff coefficient, surface temperature and precipitation). The potential runoff coefficient was estimated using the WetSpa (Water and Energy Transfer between Soil, Plants and Atmosphere) extension model (Liu and De Smedt, 2004) which incorporates three types of data inputs including land use and land cover, slope and soil texture classes as synthesized by a Table 1 (Liu and De Smedt, 2004). The soil texture and slope rasters were resampled to the same spatial resolution (300 m) of LULC maps and initial

resolution of potential runoff coefficient, which is also resampled at 0.5° resolution (the original resolution of the temperature and precipitation datasets) using the zonal statistic method. The potential runoff coefficient for impervious surfaces (IMP was estimated using Eq. (3) (Liu and De Smedt, 2004).

$$C_u = IMP + (1 - IMP) C_{grass} \tag{3}$$

where, $C_u$ is the potential runoff coefficient for built-up areas, $C_{grass}$ is the potential runoff coefficient for grassland, and IMP

is 0.50 that presents the percentage of impervious surfaces recommended for built-up zones.



The LULC maps (Figure 4) used in this study were reclassified from time series of annual global Climate Change Initiative Land Cover (CCI-LC) maps at 300 m spatial resolution covering a period of 24 years (1992 – 2015) (ESA-CCI, 2017).

**Figure 4.** A time series of annual land cover maps of Africa with 7 classes (1992 – 2015).



These land cover maps were originally classified from the landcover imagery captured by five different satellites, including the Advanced Very High-Resolution Radiometer (AVHRR), Medium Resolution Imaging Spectrometer Full Resolution and Reduced Resolution (MERIS FR and RR), SPOT-Vegetation (SPOT-VGT), Project for On-Board Autonomy, with the V standing for Vegetation (PROBA-V), Environmental Satellite-Advanced Synthetic Aperture Radar (ENVISAT-ASAR). The CCI-LC map 2015 was validated using the GlobCover map 2009 with two overall accuracy levels of 71.45% and 75.4% (ESA-CCI, 2017). Based on the CCI-LC product manual version 2.0 (ESA-CCI, 2017), all 24 CCI-LC maps were reclassified from the LCCS (Land Cover Classification System) legend to IPCC (Intergovernmental Panel on Climate Change) legend (Penman et al., 2003) that is consistent with a Table 1 of the Wetspa's potential runoff coefficient (Liu and De Smedt, 2004).

**Table 1.** Potential runoff coefficient for different land use and land cover types, slope and soil texture classes (Liu and De Smedt, 2004). Sa: Sand, LoSa: Loamy sand, SaLo: Sandy loam, Lo: Loam, SiLo: Silty loam, Si: Silt, SaClLo: Sandy clay loam, ClLo: Clay loam, SiClLo: Silty clay loam, SaCl: Sandy clay, SiCl: Silty clay, Cl: Clay.

| Land use | Slope (%) | Sa | LoSa | SaLo | Lo | SiLo | Si | SaClLo | ClLo | SiClLo | SaCl | SiCl | Cl |
|---|---|---|---|---|---|---|---|---|---|---|---|---|---|
| Forestland | < 0.5 | 0.03 | 0.07 | 0.10 | 0.13 | 0.17 | 0.20 | 0.23 | 0.27 | 0.30 | 0.33 | 0.37 | 0.40 |
| | 0.5 - 5 | 0.07 | 0.11 | 0.14 | 0.17 | 0.21 | 0.24 | 0.27 | 0.31 | 0.34 | 0.37 | 0.41 | 0.44 |
| | 5 - 10 | 0.13 | 0.17 | 0.20 | 0.23 | 0.27 | 0.30 | 0.33 | 0.37 | 0.40 | 0.43 | 0.47 | 0.50 |
| | > 10 | 0.25 | 0.29 | 0.32 | 0.35 | 0.39 | 0.42 | 0.45 | 0.49 | 0.52 | 0.55 | 0.59 | 0.62 |
| Grassland | < 0.5 | 0.13 | 0.17 | 0.20 | 0.23 | 0.27 | 0.30 | 0.33 | 0.37 | 0.40 | 0.43 | 0.47 | 0.50 |
| | 0.5 - 5 | 0.17 | 0.21 | 0.24 | 0.27 | 0.31 | 0.34 | 0.37 | 0.41 | 0.44 | 0.47 | 0.51 | 0.54 |
| | 5 - 10 | 0.23 | 0.27 | 0.30 | 0.33 | 0.37 | 0.40 | 0.43 | 0.47 | 0.50 | 0.53 | 0.57 | 0.60 |
| | > 10 | 0.35 | 0.39 | 0.42 | 0.45 | 0.49 | 0.52 | 0.55 | 0.59 | 0.62 | 0.65 | 0.69 | 0.72 |
| Croplands | < 0.5 | 0.23 | 0.27 | 0.30 | 0.33 | 0.37 | 0.40 | 0.43 | 0.47 | 0.50 | 0.53 | 0.57 | 0.60 |
| | 0.5 - 5 | 0.27 | 0.31 | 0.34 | 0.37 | 0.41 | 0.44 | 0.47 | 0.51 | 0.54 | 0.57 | 0.61 | 0.64 |
| | 5 - 10 | 0.33 | 0.37 | 0.40 | 0.43 | 0.47 | 0.50 | 0.53 | 0.57 | 0.60 | 0.63 | 0.67 | 0.70 |
| | > 10 | 0.45 | 0.49 | 0.52 | 0.55 | 0.59 | 0.62 | 0.65 | 0.69 | 0.72 | 0.75 | 0.79 | 0.82 |
| Bare lands | < 0.5 | 0.33 | 0.37 | 0.40 | 0.43 | 0.47 | 0.50 | 0.53 | 0.57 | 0.60 | 0.63 | 0.67 | 0.70 |
| | 0.5 - 5 | 0.37 | 0.41 | 0.44 | 0.47 | 0.51 | 0.54 | 0.57 | 0.61 | 0.64 | 0.67 | 0.71 | 0.74 |
| | 5 - 10 | 0.43 | 0.47 | 0.50 | 0.53 | 0.57 | 0.60 | 0.63 | 0.67 | 0.70 | 0.73 | 0.77 | 0.80 |
| | > 10 | 0.55 | 0.59 | 0.62 | 0.65 | 0.69 | 0.72 | 0.75 | 0.79 | 0.82 | 0.85 | 0.89 | 0.92 |
| Built-up areas | < 0.5 | 0.57 | 0.59 | 0.60 | 0.62 | 0.64 | 0.65 | 0.67 | 0.69 | 0.70 | 0.72 | 0.74 | 0.75 |
| | 0.5 - 5 | 0.59 | 0.61 | 0.62 | 0.64 | 0.66 | 0.67 | 0.69 | 0.71 | 0.72 | 0.74 | 0.76 | 0.77 |
| | 5 - 10 | 0.62 | 0.64 | 0.65 | 0.67 | 0.69 | 0.70 | 0.72 | 0.74 | 0.75 | 0.77 | 0.79 | 0.80 |
| | > 10 | 0.68 | 0.70 | 0.71 | 0.73 | 0.75 | 0.76 | 0.78 | 0.80 | 0.81 | 0.83 | 0.85 | 0.86 |
| Wetlands & Water | | 1.00 | 1.00 | 1.00 | 1.00 | 1.00 | 1.00 | 1.00 | 1.00 | 1.00 | 1.00 | 1.00 | 1.00 |

Referring to the soil textural triangle developed by the United States Department of Agriculture (USDA) (Fernandez-Illescas et al., 2001), the soil texture dataset at 300 m resolution (Figure 5) was estimated based on sand, clay, and silt fractions



available at 250 m resolution from the Africa Soil Information Service (AfSIS) (Hengl et al., 2015). Because the AfSIS data have gaps over the Sahara desert, in this region the soil texture was classified from the WorldGrids' s sand, clay, and silt fractions available at 1 km spatial resolution (Hengl et al., 2014). The slope map at 300 m resolution (Figure 5) was established from the Global Multi-resolution Terrain Elevation Data 2010 (GMTED2010) available at 250 m spatial resolution of the U.S.

5    Geological Survey (USGS) and the National Geospatial-Intelligence Agency (NGA) (Danielson and Gesch, 2011).

**Figure 5.** Maps of sand, clay, and silt fractions, soil texture and slope of Africa.

The long-term monthly land-surface temperature (Figure 6) was calculated from the 4.01 release of the CRU TS (Climatic Research Unit Timeseries) dataset spanning a period of 116 years (1901 – 2016) (Harris et al., 2014). This dataset

10   was developed, subsequently updated, improved and maintained with support from a number of funders, principally by the UK's Natural Environment Research Council (NERC) and the US Department of Energy. Long-term support is currently provided by the UK National Centre for Atmospheric Science (NCAS), a NERC collaborative center (Harris et al., 2014).







**Figure 6.** Maps for runoff controlling factors: potential runoff coefficient ($C_o$), surface temperature (T) and precipitation (P) utilized to establish the $C_o$, T and P overlay intersections for the interpolation guidance of the observed runoff coefficient ($R_c$).



### 2.2.3 Estimation of runoff and annual evapotranspiration

The long-term runoff depth was estimated using Eq. (4).

$$R = R_c \times P \tag{4}$$

where, R is a runoff depth (mm·month$^{-1}$), $R_c$ is a runoff coefficient (dimensionless), and P is the precipitation intensity

(mm·month$^{-1}$). The long-term annual $R_c$ is the average of all monthly long-term $R_c$ for the period starting from 1901 to 2017.

The current study estimated the long-term annual mean evapotranspiration according to the principle of water budget expressed by Eq. (5) referring to the concept of water balance, where ground water changes is approximately zero on the long-term annual period basis due to the patterns of wet and dry seasons of the year (Edwards et al., 2015).

$$E_T = P - R \tag{5}$$

where, $E_T$ is a long-term mean evapotranspiration in mm·yr$^{-1}$, P is a long-term mean precipitation in mm·yr$^{-1}$, and R is a long-term mean runoff depth in mm·yr$^{-1}$. Thus, the evapotranspiration coefficient ($ET_c$) can be expressed as the ratio of evapotranspiration to precipitation intensity received within the same basin and the identical period (Yang et al., 2018) as presented by Eq. (6).

$$ET_c = \frac{E_T}{P} \tag{6}$$

## 3 Results

Figure 7 presents the resultant maps of long-term mean monthly and annual runoff coefficient, precipitation, and runoff at 0.5° spatial resolution for the period starting from 1901 to 2017 utilized to produce the zonal statistics at continental level, within 25 major basins and 55 countries of Africa.















**Figure 7.** Maps of long-term mean monthly and annual runoff coefficient, precipitation and runoff (1901 – 2017).



### 3.1 Precipitation-runoff relationship over the continent of Africa

The zonal statistical analysis at continental level indicates that the runoff (105.72 mm·yr$^{-1}$) counts 16% of the long-term mean precipitation (672.52 mm·yr$^{-1}$) and evapotranspiration (566.80 mm·yr$^{-1}$) comprises the remaining 84% (Figure 8).

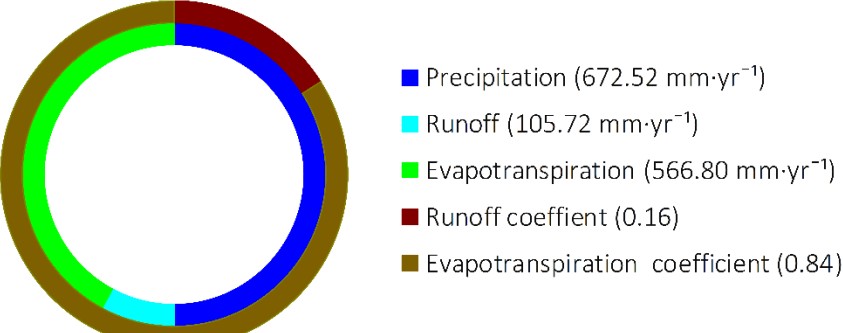

5          **Figure 8.** Long-term average annual water balance of Africa (1901 – 2017).

Assessment of the long-term monthly rainfall-runoff relationship revealed that the continent of Africa had the highest and lowest long-term mean precipitation intensities of 70.33 mm·month$^{-1}$ and 44.15 mm·month$^{-1}$ in August and June, respectively. The greatest and smallest long-term mean runoff depths of 11.16 mm·month$^{-1}$ and 5.64 mm·month$^{-1}$ are observed in October and June, respectively. The greatest rainfall-runoff correlation is noticed in October with a $R_c = 0.2$. While, the lowest $R_c = 0.1$ are recorded in July whenever $P = 57.36$ mm·month$^{-1}$ and $R = 6.3$ mm·month$^{-1}$ (Fig. 9). The long-term monthly mean observed runoff and runoff coefficient had better agreements with interpolated observed ones. Some minor mismatches observed in May, June, August, September, October and November are due to different mean precipitation estimates from two distinct zones (monitored area and whole continent) (Figure 9).

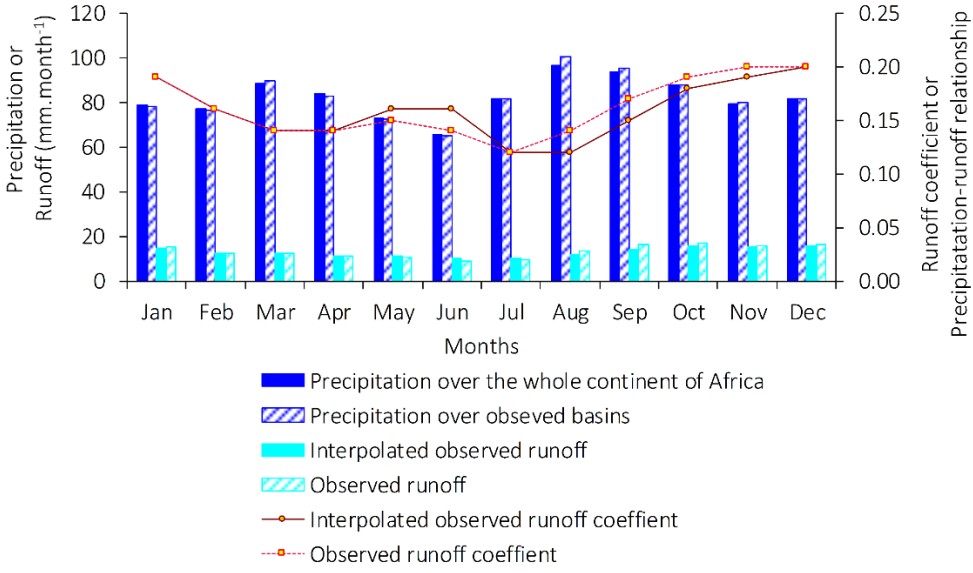

15          **Figure 9.** Long-term monthly precipitation-runoff relationship over the African continent (1901 – 2017).





## 3.2 Precipitation-runoff relationship within 25 major African basins

Figure 10 compares the long-term mean monthly and annual precipitation, interpolated observed runoff, interpolated observed runoff coefficient and long-term mean annual evapotranspiration within 25 major African basins (1901 – 2017).

**Figure 10.** Precipitation-runoff relationship within 25 major basins of Africa (1901 – 2017).



Top eight tropical basins out of 25 major African basins that comprised the highest runoff depths > 100 mm·yr$^{-1}$ are: Madagascar (575.42 mm·yr$^{-1}$), Gulf of Guinea (573.65 mm·yr$^{-1}$), Congo (355.02 mm·yr$^{-1}$), Africa-West Coast (227.68 mm·yr$^{-1}$), Africa-Indian Ocean Coast (143.89 mm·yr$^{-1}$), Angola-Coast (132.34 mm·yr$^{-1}$), Africa-East Central Coast (122.88 mm·yr$^{-1}$), and Rift Valley (101.95 mm·yr$^{-1}$). These basins also have the highest long-term annual rainfall intensities among others,

ranging from 811.88 mm·yr$^{-1}$ to 1,594.88 mm·yr$^{-1}$, and are amongst the top ten basins with the strongest correlation between rainfall and runoff compared to others with a mean runoff coefficient ranging from 0.13 to 0.39. The basins with weak precipitation-runoff relationship indicates higher $E_T$ proportions. Figure 10 also illustrates the details about monthly precipitation-runoff relationship within 25 major basins of Africa.

### 3.3 Precipitation-runoff relationship within 55 African countries

Figure 11 correlates the long-term mean monthly and annual precipitation (P), interpolated observed runoff (R), interpolated observed runoff coefficient ($R_c$) and long-term mean annual evapotranspiration ($E_T$) during 1901 – 2017 in 55 countries of Africa. P ranges from 21.38 mm·yr$^{-1}$ to 2,820.92 mm·yr$^{-1}$; R ranges from 1.06 mm·yr$^{-1}$ to 1,410.68 mm·yr$^{-1}$, in Egypt and Mauritius, respectively; $R_c$ ranges from 0.01 in Gambia to 0.5 in Mauritius; $E_T$ ranges from 20.32 mm·yr$^{-1}$ in Egypt to 2,099.14 mm·yr$^{-1}$ in Sierra Leone. It should be noticed that, with the highest runoff depths > 300 mm·yr$^{-1}$, the top 12

countries, including Mauritius, Equatorial Guinea, Gabon, Sierra Leone, Liberia, Madagascar, São Tomé and Príncipe, Republic of Congo, Cameroon, D. R. Congo, Seychelles, and Guinea-Bissau are ranked among the top 20 out of 50 countries that experiences the greatest rainfall-runoff correlation with runoff coefficients range from 0.17 to 0.50. For comparative illustration of the long-term average monthly precipitation, runoff and runoff coefficient between 55 countries of Africa, see Figure 11.






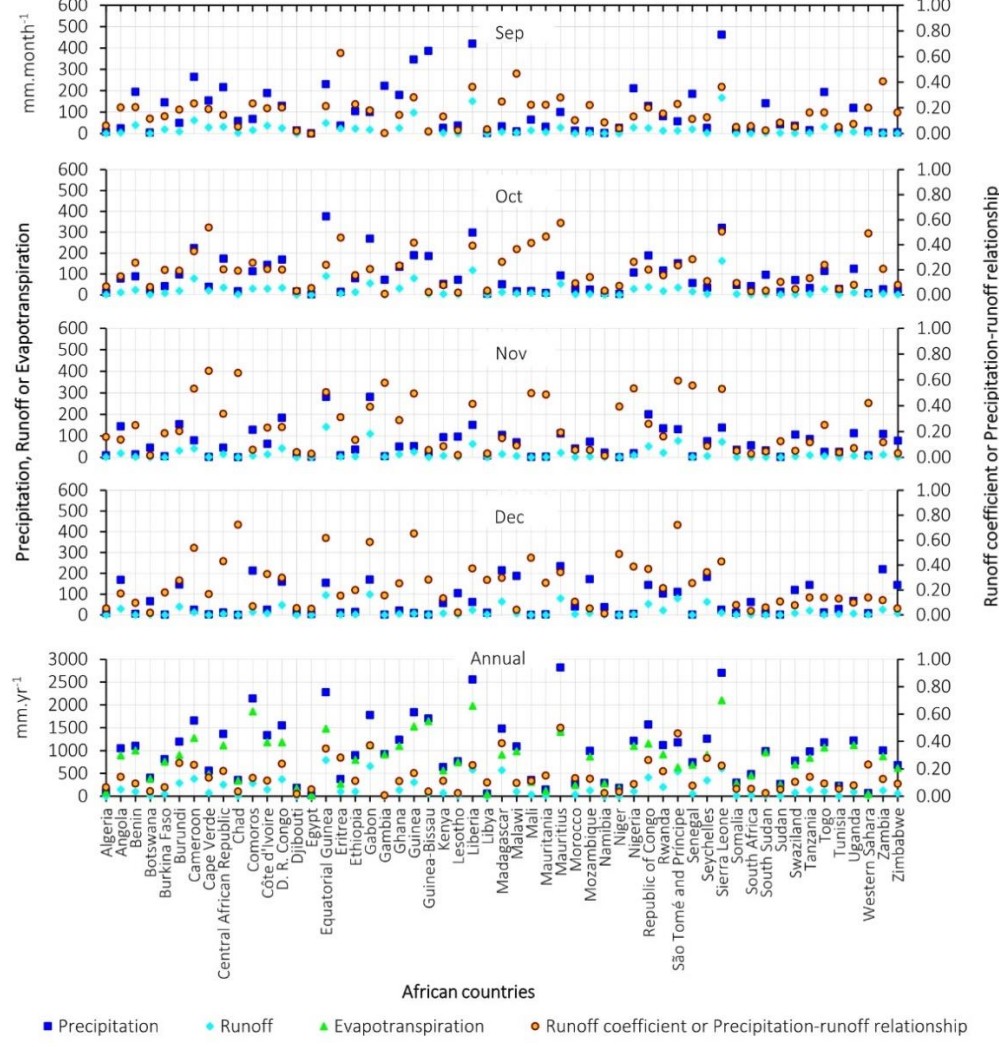

**Figure 11.** Precipitation-runoff relationship within 55 countries of Africa (1901 – 2017).

## 4 Discussions

5        Streamflow and rain gauging stations are known as a trustable source of reliable data for different hydrological studies (Urroz et al., 2001). The GRDC river discharges and GPCC precipitation datasets have outreaching accuracy for spatial analysis of the precipitation-runoff relationship rather than relying solely on the runoff model-based estimates which are likely associated with huge uncertainties caused by non-error free data and sometimes un-well-constructed models (Loucks et al., 2005). The study conducted by the University of New Hampshire-Global Runoff Data Centre (UNH-GRDC): "high-resolution

10  fields of global runoff combining observed river discharge and simulated water balances" that has been considered as reference to validate the runoff-related hydrological studies since the beginning of 21$^{st}$ century (Fekete et al., 2002; Hong et al., 2007)



was compared with the current study based on the latitudinal zones at 1° interval scale (Figure 12). This analysis indicates that the long-term mean annual mean rainfall (1920 – 1980) version 2.01 (Willmott et al., 1998; Fekete et al., 2002) that was utilized to simulate the UNH/GRDC composite runoff (Fekete et al., 2002) is roughly matching with the long-term (1901 – 2017) mean annual GPCC precipitation estimated in the current study (Figure 12). Our comparative analysis also shows better

agreements over the northern hemisphere between 36ºN and 14ºN, in the southern hemisphere between 17ºS and 34ºS latitudes, and in the equatorial zone laying between 4ºN and 8ºS. Major differences are between 14ºN and 4ºN in the northern hemisphere and in the southern hemisphere between 8ºS and 17ºS latitudes. These differences are possibly due to the UNH/GRDC method that assigned the same runoff depths in observed and unobserved basins that led to overestimation of the runoff in drylands of Australia and Africa (Fekete et al., 2002). It should be noted that the interpolation method of observed runoff coefficient under

consideration of environmental characteristics (land use and cover types, slope, soil texture classes, surface temperature and precipitation) has improved of both runoff coefficient and runoff over unobserved African regions rather than assuming similar runoff depths to different catchments with different environmental characteristics.

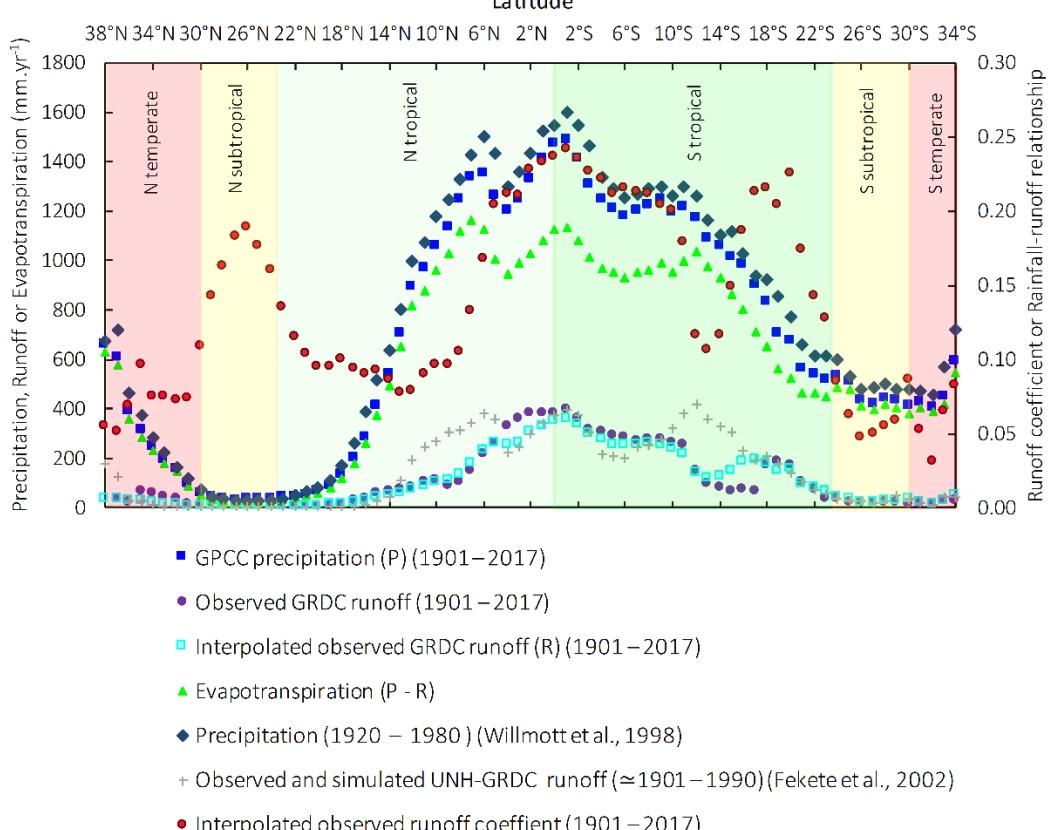

**Figure 12.** Comparison of the precipitation-runoff relationship between the UNH-GRDC (Fekete et al., 2002) and the present study.

The latitudinal profile analysis revealed that 1ºS latitude is the most hotspot region of Africa for both high mean annual rainfall intensity (1,486 mm·yr⁻¹), and runoff depth (357.26 mm·yr⁻¹), followed by the equatorial zone at 0º latitude with a mean annual



rainfall intensity of 1,466.46 mm·yr$^{-1}$, runoff depth of 346.66 mm·yr$^{-1}$, highest runoff coefficient value of 0.24 for both 1°S and equatorial zone (0° latitude). These latitudinal zones also comprised of the highest evapotranspiration rates ranging between 1,119.8 mm·yr$^{-1}$ and 1,157.24 mm·yr$^{-1}$. This might be one of the remarkable proofs showing good estimates of the present study based on the well-known distribution of precipitation across different latitudinal climatic zones (Peel et al., 2007).

Based on the following six latitudinal climate zones: northern (N) tropical (0°Equator ≤ latitude ≤ 23.4°N), southern (S) tropical (23.4°S ≤ latitude < 0°Equator), N subtropical (23.4°N ≤ latitude ≤ 30°N), S subtropical (23.4°S ≤ latitude ≤ 30°S), N temperate (30°N < latitude ≤ 72°N), and S temperate (30°S < latitude ≤ 72°S), the long-term annual mean of the water balance's variables, including precipitation (P), evapotranspiration (E$_T$), interpolated observed runoff (R) and their corresponding interpolated observed runoff coefficient (R$_c$) for the period of 117 years (1901 – 2017) were estimated and

presented in Table 2 to provide tangible statistics corresponding to Figure 12.

**Table 2.** Long-term annual water balance and runoff coefficient within the African latitudinal climatic zones (1901 – 2017).

| Latitudinal climatic zones | % area Africa | P (mm·yr$^{-1}$) | E$_T$ (mm·yr$^{-1}$) | R (mm·yr$^{-1}$) | R$_c$ |
|---|---|---|---|---|---|
| **Tropical** | **75** | **835** | **699** | **136** | **0.16** |
| N tropical | 48 | 703 | 607 | 96 | 0.14 |
| S tropical | 26 | 1,076 | 865 | 211 | 0.20 |
| **Subtropics** | **17** | **156** | **143** | **13** | **0.08** |
| N subtropical | 13 | 37 | 31 | 6 | 0.16 |
| S subtropical | 5 | 468 | 437 | 31 | 0.07 |
| **Temperate zones** | **8** | **261** | **243** | **18** | **0.07** |
| N temperate | 6 | 210 | 195 | 15 | 0.07 |
| S temperate | 2 | 432 | 406 | 26 | 0.06 |

      Because of the Saharan desert located in northern Africa, the northern tropical and temperate zones have lower mean annual precipitation and runoff compared to southern tropical and temperate zones. Compared to the tropical zone, subtropics and temperate zones of Africa had low rainfall and runoff amounts (Figure 12 and Table 2), which expose them to the water

scarcity problem and less rainfall-runoff related disasters. The problem of insufficient water resource in drylands and semi-humid regions can be managed through the use of groundwater, minimization of water losses, establishment of dams, inter-basins water transfer, wastewater reuse, and surface water desalination instead of relying on unmodernized water supply technologies that are no longer meeting the water demand for our contemporary development associated with fast population growth (Maliva and Missimer, 2012). The tropical zone comprises high precipitation intensity which produces huge runoff

volumes enough for underground and surface water replenishment. While, excessive surface waterflow induces significant damages which require adequate strategies by promoting practical stormwater management systems (e.g.: forestation, different types of terraces and dams, etc.) that have potential ability to curb the movement and effects of surface water runoffs (Ponette-González et al., 2015; Karamage et al., 2017b).



## 5 Conclusions

The present study investigated the spatial relationship between precipitation and runoff using the runoff coefficient as the measurement indicator, estimated from the long-term monthly runoff calculated based on the Global Runoff Data Centre (GRDC)'s streamflow records and Global Precipitation Climatology Centre (GPCC) rainfall data for a temporal period of 117 years (1901 – 2017) within monitored basins covering ≃ 41.4% of the total African continent. The interpolation method of observed runoff coefficient directed by the ancillary data (potential runoff coefficient, land-surface temperature, and precipitation) that affect the runoff generation process has improved the estimation of runoff coefficient and runoff depths in ungauged basins. Thus, this study provides insightful hydrological information on the precipitation-runoff relationship at three spatial scales, including the whole continent, 25 major basins and 55 African countries which could raise the awareness to a wide range of relevant stakeholders. This study also suggested that the problem of water scarcity in drylands and semi-humid region can be handled through the use of groundwater, minimization of water losses, establishment of dams, inter-basins water transfer, wastewater reuse, and water desalination technologies. The tropical stormwater-runoffs require suitable water management programs such as for example forestation, establishment of different types of terraces and dams, etc. in order to minimize the waterflow-related disasters.

**Acknowledgments:** Many thanks to the editor for the insightful comments that helped us to improve the quality of this manuscript. This study was supported by: (a) the Chinese Academy of Sciences and the World Academy of Sciences (CAS-TWAS) President's PhD Fellowship Program, and (b) the Sino-Africa Joint Research Centre, Chinese Academy of Sciences (No. SAJC201609).

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
