# Peer review of "Spatial Relationship between Precipitation and Runoff in Africa"

_Hydrology and Earth System Sciences, 2018_

## Referee Comment (RC1) · Anonymous Referee #1 · 18 Oct 2018

I would like to start my comments by referring to one of the authors conclusions that states 'The interpolation method of observed runoff coefficient ...... that affect the runoff generation process has improved the estimation of runoff coefficient and runoff depths in ungauged basins'. This is simply NOT TRUE for several reasons. The most obvious reason is that the paper includes no validation of the estimated runoff coefficients. The 2nd reason is that the runoff data set used to establish the runoff coefficient estimates is based on observed data from some basins which have huge impacts of reservoir storage, hydropower releases, irrigation abstractions (and others) all of which will affect either the annual runoff coefficient and/or the seasonal patterns of runoff coefficient. The 3rd reason is that many of the observed runoff data represent very large catchments that have hugely spatially variable patterns of runoff such that an average runoff

coefficient would be meaningless. The 4th reason is that in many parts of the continent runoff coefficients will be strongly related to topographic characteritics that might not be adequately reflected in the input variables used by the authors. This is not the first paper that attempts to apply methods at very course spatial scales and to suggest (without any validation what so ever) that the outputs will be useful to water resources management. Quite often these papers (as does this one) criticise the use of 'un-well-constructed models' (page 21) that are based on non-error free data. Are the authors seriously suggesting that their data are error-free, because this is a claim that can very easily be refuted. There are many people within the African continent (and from other countries) who have been using hydrological and water resources assessment models for practical water resources management and are unlikely to see the results of this study as adding anything, either from a scientific or practical persepective, to the approaches that can be applied. Apart from the points that I have already raised about the complete lack of validation, the spatial scale of the study is simply too coarse to be of any value to the type of water resources management and planning issues that confron African countries.

I also found it rather interesting that the authors fail to qoute any of the scientific literature that has been produced in the region on the subject of water resources estimation (see the rather condescending sentence on line 10 of page 2).

In summary, this study is seriously flawed from a scientific hydrology perspective and adds nothing to either African hydrological sciences nor to the methods that can be used to manage water resources over different parts of the continent.
* * *

---

## Short Comment (SC1) · 19 Oct 2018

I will start by appreciating the amount of energy put in gathering, processing and analyzing the huge amount of hydrological data on Africa. However, in agreement with Referee #1, whose contributions are already posted, I noted a number of unfounded pronouncements. I will not repeat on issues already raised by Referee #1. My comments/remarks are summarized as follows:

- It is common knowledge that there are many factors with significant control on the hydrology of catchments/basins; therefore, it is always important for authors to justify their choice of factors to include/exclude in their study. The authors listed their choices on page 2, lines 24-26, without any attempt to explain why they considered them to be

the more important ones.

- The choice of a country as the finest spatial scale in the study also required a base. It is not clear how the authors can justify their choice and how they treated the fact that many African countries drain into more than one river basin. This is important because it might affect the results shown in Figure 11.

- The results of the current study seem to confirm what is already known about the hydrology of Africa, which leaves one wondering whether the authors were able to articulate the objectives of the study. The discussion part of the paper did not help to put the study into proper context either.

- The immediate above remark, perhaps, explain the weak discussion and, especially, the conclusion of the paper. For example, the study has no basis for concluding that items enumerated on page 24, lines 11-12, are the solutions to address water scarcity in Africa.

My opinion is that, while a lot of effort was put in gathering, processing and analyzing the data, there was not much zeal in interpreting and interrogating the results leading to a weak discussion of the results and conclusions.
* * *

---

## Referee Comment (RC2) · Anonymous Referee #2 · 18 Nov 2018

I reviewed the manuscript "spatial relationship between precipitation and runoff in Africa", and I feel the manuscript has some major flaws. Besides the fact that the authors use a data set which has significant data gaps which are not discussed in the paper. This may have major implications for the validity of the results. Also the fact that the river basins change over time (eg anthropological changes) which are not taken into consideration. The approach taken to generate information in ungauged basins is not sufficiently validated with the observed data to be able to use them for the final analyses. Also the approach is not sufficiently described (eg what is the difference between the potential runoff coefficient and the final one), how are T and P used to generate the Rc? Below are additional general comments on the manuscript, also the paper does not use common hydrological terminology, which I have highlighted in the

last section with detailed comments.

General comments:

Page 2, Line 10 "Although runoff studies have been conducted at global scale and in some local areas in Africa" Although there are few studies describing continental scale hydrology in Africa (but they do exist, see Schuol et al 2008), it is pertinent untrue to say there are only some studies on runoff in local areas in Africa. The authors are advised to perform a detailed literature review before writing the introduction and providing such an untrue statement.

Page 2, Line 24-26 "The Rc data were then interpolated to the ungauged areas using the key factors such as land-surface temperature (T), Precipitation (P) and potential runoff coefficient (Co) estimated from the land use and land cover, soil texture and slope information by using GIS spatial analysis techniques." This is a typical approach to regionalise hydrological processes in ungauged basins. The authors should refer to earlier studies under the PUB initiative of the IAHS to explain how this approach is a well accepted approach. However, in this approach I don't understand how the runoff coefficient (Rc) is dependent on the runoff coefficient (Co)? Are these not the same thing?

Page 5 figure 2, the conceptual framework calculates the runoff coefficient from the rainfall and runoff data bases, afterwards it recalculates the runoff by multiplying the rainfall with the runoff coefficient (top right box), how is this relevant?

Page 6, figure 3, colors indicated in the legend are not consistent with the map, unclear what the legend "mean number of streamflow recorded months per each month of the year during 1901-2017" means. Based on the text, this would mean 12 maps? Also I see some areas with yellow markers indicating less than 10 years of data, correct? And for the Congo river I only see a marker at the downstream end of the basin. How does this affect the analyses? Also using data from any period seriously affects the analyses, as many dams have been constructed in the later decades. Even for stations with data

across the 117 years, this needs to be taken into consideration.

Page 6 equation 1 is an obvious equation and does not need to be presented.

Page 7 line 1-10: using two different datasets may bring in additional uncertainty, is it really worth including the additional 3 years of data?

Page 7 equation 2, does this mean you have for each station 12 Rc values for each month? The units for runoff and precipitation (line 14) should be mm/month. How do you convert this to an Rc on an annual basis? How do you take into account different availability of data for specific months?

Section 2.2.1: how many stations were used for the study with a typical availability of data?

Section 2.2.2: why is there a completely different approach for ungauged basins? How are the two studies linked?

Page 7 line 20: the authors are using IDW to interpolate runoff coefficients. However Rc is dependent on the upstream catchment area, two stations close to each other could have significant differences in upstream catchment area and dynamics, this is not an approach which has been tested in the hydrological field (neither are the authors providing evidence that this approach is appropriate).

page 11&12 figure 6: how was the most right map developed from the three other maps (not explained in text) What would be interesting is to assess how the approach in 2.2.2 is able to generate Rc for the gauged basins to validate the approach used.

Page 13 line 11 it is very confusing when the authors use ETc in a different way it is normally used (for crop evapotranspriration)

Page 13,14,15 section 3: there is absolutely no reflection on what figure 6&7 are showing, is this the result of the methodology described in section 2.2.1 or 2.2.2? How does the interpolation approach work compared to the one using observed data?

Page 17, section 3.1 to base any conclusion on the validity of the approach solely on continental scale data, is problematic. There is huge spatial and temporal variability across the continent, average monthly rainfall as presented in this section is irrelevant. Also observed runoff presented in figure 9 does not include the entire continent, how can this be compared to the interpolated one which covers the entire continent? The observed basins often collect data from large river basins, which have different dominating processes compared to smaller basins.

Page 19, section 3.3 To estimate runoff coefficients on country scale is irrelevant, as they do not constitute a drainage basin.

Schuol, J., Abbaspour, K.C., Yang, H., Srinivasan, R. and Zehnder, A.J., 2008. Modeling blue and green water availability in Africa. Water Resources Research, 44(7).

Detailed comments:

Use of definitions: Page 1 Line 28 "lacking precipitation"

Line 30 "precipitation scarcity"

Line 31 "eastern and western drylands of Africa"

Page 2 Line 4 "water runoffs"

Line 4 "hazards and disasters"

Line 5-6 "flood threats"

Line 12 "indicative runoff coefficient" what do you mean with indicative?

Line 16 "The runoff coefficient is the ratio of runoff depth to rainfall intensity" rainfall intensity is more often associated with events and not with long term runoff coefficient as you probably are referring to.

Line 17 "waterflow"

Line 19 "runoff coefficient is very useful for catchment scale land use and flood management", I disagree with this statement, floods are often associated with short timespans

Line 30-31 "Evapotranspiration is generally less than precipitation in wet seasons, that is positive water balance due to groundwater accumulation, which results in an increased surface runoff." I do not understand this sentence

Line 32 "plants absorb underground water" why only the plants?

Line 33-34 "underground water can be ignored in the long-term annual mean water balance" Do you mean that "the change in storage" can be ignored?

Page 3 Line 8 unit for mean precipitation should be mm/year

Source of data for figure 1?

Page 4 Line 12-13 "Drainage patterns are controlled by the distribution of basins and swells, about 95% is drained through permanent or ephemeral rivers" what other types of rivers exist? And what is a swell? Aren't drainage patterns dependent on geographical location, topography, climatological factors etc?

Line 14 "sand sea"??

---

## Author Comment (AC1) · 27 Feb 2019

RC1-Anonymous Referee #1

Anonymous Referee of the HESS Journal

Dear Anonymous Referee,

Subject: Responses for your review comments posted on 18 October 2018 on our manuscript No.: hess-2018-424, entitled "Spatial Relationship between Precipitation and Runoff in Africa"

We would like to thank you for the time and effort used to review our manuscript. We have carefully reviewed the comments and have revised the manuscript accordingly.

Our responses are given point by point below and the track-change and clean-revised manuscripts were also prepared. We thankfully acknowledge your comments, as they were valuable in improving the quality of our manuscript and are useful in our future work.

Yours sincerely,

Review comment 1

I would like to start my comments by referring to one of the authors conclusions that states 'The interpolation method of observed runoff coefficient ...... that affect the runoff generation process has improved the estimation of runoff coefficient and runoff depths in ungauged basins'. This is simply NOT TRUE for several reasons. The most obvious reason is that the paper includes no validation of the estimated runoff coefficients.

Response

We would like to apologize for the inconsistent statements, mistakes, and unwell-described concepts. Based on your comments was majorly revised and more details were provided. In the revised manuscript, this study has 2 objectives: (1) the estimation of the relationship between precipitation and runoff using the runoff discharges downscaled from basin to grid-scale which can be reasonably utilized on the non-catchment regional studies (i.e.: Country scale), (2) prediction of runoff depths and coefficients over ungauged regions utilizing the inter-gauged and ungauged basin parameter transfer method based on spatial hydrologic similarities. This is one of the recommended approaches for hydrological predictions in ungauged basins (PUB) (Bárdossy, 2007; Blöschl, 2006; Chiew et al., 2018). This method assumes that two separate catchments can have a similar hydrological process if they have similar climatic and physical conditions. Hydrologic similarity was assessed based the key runoff controlling factors, including antecedent soil moisture condition (AMC), Natural Resources Conservation Service (NRCS) runoff curve number (CN), terrestrial water storage change (TWSC), surface temperature (T), and topographic parameters (topographic wetness

index (TWI) and slope). Regarding the validation of the approach used to predict the data for filling the gaps indicated that the estimated and observed runoff coefficients have the goodness of fit ($R^2$) ranging from 0.56 to 0.67 for the long-term monthly Rc and 0.78 for the annual mean Rc (Figure 14). These results are within permissible validity limits since an $R^2 > 0.5$ is considered acceptable for calibration and validation in hydrological modeling (Santhi et al., 2001; Van Liew et al., 2003). It can be concluded that inter-gauged and ungauged basin parameter transfer based on hydrologic similarity is an alternative approach for gaps filling in runoff prediction and it can even perform much better if the input observed runoff discharges do not have a lot of temporal gaps.

Review comment 2

The 2nd reason is that the runoff data set used to establish the runoff coefficient estimates is based on observed data from some basins which have huge impacts of reservoir storage, hydropower releases, irrigation abstractions (and others) all of which will affect either the annual runoff coefficient and/or the seasonal patterns of runoff coefficient.

Response

Thank for your valuable comment about water storage changes. In our revised manuscript monthly water storage change within different parts of the continent were considered in the hydrologic similarity analysis (Figure 7) using the terrestrial water storage changes estimated from the Center for Space Research (CSR) Gravity Recovery and Climate Experiment (GRACE) RL05 mascon solutions available at 1o resolution for the period starting from April 2002 to June 2016 (Save et al., 2016). Except, the precipitation datasets available since the beginning of 20th century, even before, the other above-mentioned changing runoff controllers are available for the recent decades (i.e.: GRACE data for water storage change analysis were collected since 2002 and good quality land cover maps are available since the 1990s). Lack of these data for the earlier decades constrained us to predict the past runoff process. Again, if the earlier runoff discharges are excluded from the long-term runoff calculations, spatial gaps would be increased and bring more challenge for validation.

Review comment 3

The 3rd reason is that many of the observed runoff data represent very large catchments that have hugely spatially variable patterns of runoff such that an average runoff coefficient would be meaningless.

Response

In the revised manuscript, runoff discharges for very large catchments were replaced by the sub-catchments with the medium size. In addition, using the Natural Resources Conservation Service (NRCS) runoff curve number (CN), the basin's runoff discharge was downscaled at a grid scale which can be reasonably utilized on the non-catchment regional studies (i.e.: Country scale). Actually, runoff-related studies are often conducted at a drainage basin scale, but, hydrological studies on the grid and country scales are very useful at the country level since each government has own policies for water resource management. For instance, it has been noticed that runoff discharges are useful in water stress analysis at country scale (Ruess, 2015; Smakhtin, 2004). Integration of NRCS-CN in downscaling the runoff discharges do not alter the quantity of observed runoff at a catchment scale, but it redistributes catchment's discharged runoff volume to its grids proportionally according to their respective climate and physical conditions.

Review comment 4

The 4th reason is that in many parts of the continent runoff coefficients will be strongly related to topographic characteristics that might not be adequately reflected in the input variables used by the authors. This is not the first paper that attempts to apply methods at very coarse spatial scales and to suggest (without any validation what so ever) that the outputs will be useful to water resources management. Quite often these papers

(as does this one) criticize the use of 'unwell-constructed models' (page 21) that are based on non-error free data. Are the authors seriously suggesting that their data are error-free, because this is a claim that can very easily be refuted There are many people within the African continent (and from other countries) who have been using hydrological and water resources assessment models for practical water resources management and are unlikely to see the results of this study as adding anything, either from a scientific or practical perspective, to the approaches that can be applied. Apart from the points that I have already raised about the complete lack of validation, the spatial scale of the study is simply too coarse to be of any value to the type of water resources management and planning issues that confront African countries. I also found it rather interesting that the authors fail to quote any of the scientific literature that has been produced in the region on the subject of water resources estimation (see the rather condescending sentence on line 10 of page 2). In summary, this study is seriously flawed from a scientific hydrology perspective and adds nothing to either African hydrological sciences nor to the methods that can be used to manage water resources over different parts of the continent.

Response

Thank you for your concern about the topographic factor on the hydrologic process. In the revised manuscript, the following key runoff controlling factors were utilized in the hydrologic similarity analysis: antecedent soil moisture condition (AMC), Natural Resources Conservation Service (NRCS) runoff curve number (CN), terrestrial water storage change (TWSC), surface temperature (T), and topographic parameters (topographic wetness index (TWI) and slope). We would like to apologize for the above-mentioned misstatement. You are right, there are no data with error-free. Really, several hydrological models and methods have been developed and they are very useful in water resource management in different part of the world but, they have some limitations depending on the case study either due to the lack of sufficient and reliable input dataset or their development that cannot allow easy incorporation of additional parameters (Lim et al., 2006). Regarding the validation of the approach used to predict the data for filling the gaps indicated that the estimated and observed runoff coefficients have the goodness of fit (R2) ranging from 0.56 to 0.67 for the long-term monthly Rc and 0.78 for the annual mean Rc (Figure 14). These results are within permissible validity limits since an R2 > 0.5 is considered acceptable for calibration and validation in hydrological modeling (Santhi et al., 2001; Van Liew et al., 2003). It can be concluded that inter-gauged and ungauged basin parameter transfer based on hydrologic similarity is an alternative approach for gaps filling in runoff prediction and it can even perform much better if the input observed runoff discharges do not have a lot of temporal gaps. Actually, runoff-related studies are often conducted at a drainage basin scale, but, hydrological studies on the grid and country scales are also very useful at the country level since each government has own policies for water resource management. For instance, it has been noticed that runoff discharges are useful in water stress analysis at a country scale (Ruess, 2015; Smakhtin, 2004).

As a scientific contribution, this study highlighted step by step how the Natural Resources Conservation Service (NRCS) runoff curve number (CN) can be a prominent proxy for the basin's runoff discharge downscaling at a grid scale which can be reasonably utilized on the non-catchment regional studies (i.e.: Country scale). In addition, this study highlighted the performance of inter-gauged and ungauged basin parameter transfer based on hydrologic similarity over the large scale as the African continent. This method indicated to be is an alternative approach for gaps filling in runoff prediction and it can even perform much better if the input observed runoff discharges do not have a lot of temporal gaps.

Again, we are thankful and appreciate your valuable comments that very helpful in revising our manuscript.

Please also note the supplement to this comment:
https://www.hydrol-earth-syst-sci-discuss.net/hess-2018-424/hess-2018-424-AC1-

supplement.zip

---

## Author Comment (AC2) · 28 Feb 2019

SC1-Anonymous Referee #3 Anonymous Referee of the HESS Journal Dear reviewer M. Mutema,

Subject: Responses for your review comments posted on 18 October 2018 on our manuscript No.: hess-2018-424, entitled "Spatial Relationship between Precipitation and Runoff in Africa"

We would like to thank you for the time and effort used to review our manuscript. We have carefully reviewed the comments and have revised the manuscript accordingly. Our responses are given point by point below and the track-change and clean-revised manuscripts were also prepared. We thankfully acknowledge your comments, as they

were valuable in improving the quality of our manuscript and are useful in our future work.

Yours sincerely,

Review comment 1

I will start by appreciating the amount of energy put in gathering, processing and analyzing the huge amount of hydrological data on Africa. However, in agreement with Referee #1, whose contributions are already posted, I noted a number of unfounded pronouncements. I will not repeat on issues already raised by Referee #1. My comments/remarks are summarized as follows: - It is common knowledge that there are many factors with significant control on the hydrology of catchments/basins; therefore, it is always important for authors to justify their choice of factors to include/exclude in their study. The authors listed their choices on page 2, lines 24-26, without any attempt to explain why they considered them to be the more important ones.

Response

Thank you for the remarks about the choice of runoff controlling factors. Based on your comments the interpolation method was revised and more details were provided. Actually, the methodology used to predict the runoff depths and coefficients in ungauged regions is the inter-gauged and ungauged basin parameter transfer method that can be considered as a hybrid interpolation method based on spatial hydrological similarities analyzed using the key runoff controlling factors. This is one of the recommended approaches for hydrological predictions in ungauged basins (PUB) (Bárdossy, 2007; Blöschl, 2006; Chiew et al., 2018). This method assumes that two separate catchments can have a similar hydrological process if they have similar climatic and physical conditions. In the revised manuscript, this study has 2 objectives: (1) the estimation of the relationship between precipitation and runoff using the runoff discharges downscaled from basin to grid-scale which can be reasonably utilized on the non-catchment regional studies (i.e.: Country scale), (2) prediction of runoff depths and coefficients

over ungauged regions utilizing the inter-gauged and ungauged basin parameter transfer method based on spatial hydrologic similarities. This is one of the recommended approaches for hydrological predictions in ungauged basins (PUB) (Bárdossy, 2007; Blöschl, 2006; Chiew et al., 2018). This method assumes that two separate catchments can have a similar hydrological process if they have similar climatic and physical conditions. The hydrologic similarity was assessed based on the key runoff controlling factors selected based on their potential sensitivity in runoff generation process as revealed by different hydrologists. Detailed information about the choice of runoff controlling factors was provided in the revised manuscript. Regarding the validation of the approach used to predict the data for filling the gaps indicated that the estimated and observed runoff coefficients have the goodness of fit ($R^2$) ranging from 0.56 to 0.67 for the long-term monthly Rc and 0.78 for the annual mean Rc (Figure 14). These results are within permissible validity limits since an $R^2 > 0.5$ is considered acceptable for calibration and validation in hydrological modeling (Santhi et al., 2001; Van Liew et al., 2003).

Review comment 2

The choice of a country as the finest spatial scale in the study also required a base. It is not clear how the authors can justify their choice and how they treated the fact that many African countries drain into more than one river basin. This is important because it might affect the results shown in Figure 11.

Response

Thank you for raising the issue concerning the choice of a country as the finest spatial scale in the study. In the revised manuscript, the relevancy of runoff estimation on a non-catchment scale was highlighted. Actually, runoff-related studies are often conducted at a drainage basin scale, but, hydrological studies on the grid and country scales are also very useful at the national level since each government has own policies for water resource management. For instance, it has been noticed that

runoff discharges are useful in water stress analysis on a country scale (Ruess, 2015; Smakhtin, 2004). Utilization of average basin estimates directly at a country level or any other non-catchment locality seems to be unrealistic. This is the reason why this study suggested the process of downscaling the basin' observed runoff discharges based on grids' direct runoff contributions to their corresponding basins which helps to incorporate the effect of major runoff controlling factors (i.e.: land cover types, soil characteristics, moisture conditions and precipitation intensities) within different grids sharing the same catchment according to the Natural Resources Conservation Service (NRCS) runoff curve number (CN) variables.

Comment 3

The results of the current study seem to confirm what is already known about the hydrology of Africa, which leaves one wondering whether the authors were able to articulate the objectives of the study. The discussion part of the paper did not help to put the study into proper context either.

Response

We apologize for the poor articulation of the objectives and discussion of the study. In the revised manuscript, the objectives and the contribution of the study were revised accordingly. "The study has 2 objectives: (1) the estimation of the relationship between precipitation and runoff using the runoff discharges downscaled from basin to grid-scale which can be reasonably utilized on the non-catchment regional studies (i.e.: Country scale), (2) prediction of runoff depths and coefficients over ungauged regions utilizing the inter-gauged and ungauged basin parameter transfer method based on spatial hydrologic similarities."

Comment 3

The immediate above remark, perhaps, explains the weak discussion and, especially, the conclusion of the paper. For example, the study has no basis for concluding that

items enumerated on page 24, lines 11-12, are the solutions to address water scarcity in Africa. My opinion is that, while a lot of effort was put in gathering, processing and analyzing the data, there was not much zeal in interpreting and interrogating the results leading to a weak discussion of the results and conclusions.

Response

We are sorry for unwell-developed discussion and conclusion. In the revised manuscript discussion and conclusion sections were revised according to the objectives of the study.

Again, we are thankful and appreciate your valuable comments that very helpful in revising our manuscript.

Please also note the supplement to this comment:
https://www.hydrol-earth-syst-sci-discuss.net/hess-2018-424/hess-2018-424-AC2-supplement.zip

---

## Author Comment (AC3) · 28 Feb 2019

RC1-Anonymous Referee #2

Anonymous Referee of the HESS Journal

Dear Anonymous Referee,

Subject: Responses for your review comments posted on 18 November 2018 on our manuscript No.: hess-2018-424, entitled "Spatial Relationship between Precipitation and Runoff in Africa"

We would like to thank you for the time and effort used to review our manuscript. We have carefully reviewed the comments and have revised the manuscript accordingly.

[Figure]

Our responses are given point by point below and the track-change and clean-revised manuscripts were also prepared. We thankfully acknowledge your comments, as they were valuable in improving the quality of our manuscript and are useful in our future work.

Yours sincerely,

Review comment 1

I reviewed the manuscript "spatial relationship between precipitation and runoff in Africa", and I feel the manuscript has some major flaws. Besides the fact that the authors use a dataset which has significant data gaps which are not discussed in the paper. This may have major implications for the validity of the results. Also, the fact that the river basins change over time (eg anthropological changes) which are not taken into consideration. The approach taken to generate information in ungauged basins is not sufficiently validated with the observed data to be able to use them for the final analyses. Also, the approach is not sufficiently described (eg what is the difference between the potential runoff coefficient and the final one), how are T and P used to generate the Rc? Page 7 line 20: the authors are using IDW to interpolate runoff coefficients. However, Rc is dependent on the upstream catchment area, two stations close to each other could have significant differences in the upstream catchment area and dynamics, this is not an approach which has been tested in the hydrological field (neither are the authors providing evidence that this approach is appropriate). How does the interpolation approach work compared to the one using the observed data? Page 17, section 3.1 to base any conclusion on the validity of the approach solely on continental scale data, is problematic. There is huge spatial and temporal variability across the continent, average monthly rainfall as presented in this section is irrelevant.

Response

Thank you for raising the issue of data gaps and validity of the approach used to predict runoff coefficients and runoff depths in ungauged basins. Based on your comment the

interpolation method was revised and more details were provided. The objective of this study was achieved using the inter-gauged and ungauged basin parameter transfer method based on spatial hydrological similarities analyzed using the key runoff controlling factors. This is one of the recommended approaches for hydrological predictions in ungauged basins (PUB) (Bárdossy, 2007; Blöschl, 2006; Chiew, Zheng, & Potter, 2018). This method assumes that two separate catchments can have a similar hydrological process when they have the same range of climatic and physical conditions that can be assessed by key runoff controlling factors such as land use types, soil moisture condition, temperature, terrestrial water storage changes, and topographic characteristics. If one of the catchments is observed it can be a source of data to unobserved one. Regarding the validation of this method, the efficiency analysis of the approach used to predict the data for filling the gaps suggested that the estimated and observed runoff coefficients have the goodness of fit (R2) ranging from 0.56 to 0.67 for the long-term monthly Rc and 0.78 for the annual mean Rc (Figure 14). These results are within permissible validity limits since an R2 > 0.5 is considered acceptable for calibration and validation in hydrological modeling (Santhi et al., 2001; Van Liew, Arnold, & Garbrecht, 2003). It can be concluded that inter-gauged and ungauged basin parameter transfer based on hydrologic similarity is an alternative approach for gaps filling in runoff prediction and it can even perform much better if the input observed runoff discharges do not have a lot of temporal gaps. Concerning the description of the approach used to generate runoff depths and runoff coefficients over ungauged regions, more details were provided and the selection of parameter was improved and described. As a scientific contribution, this study highlighted step by step how the Natural Resources Conservation Service (NRCS) runoff curve number (CN) can be a prominent proxy for the basin's river discharge downscaling at a grid scale which can be reasonably utilized on the non-catchment regional studies (i.e.: Country scale). Integration of NRCS-CN in downscaling the runoff discharges do not alter the quantity of observed runoff at a catchment scale, but it redistributes catchment' s discharged runoff volume to its grids proportionally according to their respective climate and physical conditions. NRCS-

CN is very useful in various hydrological studies mainly in predicting the direct runoff discharges by incorporating the land use and land cover (LULC) information, soil hydrological characteristics, antecedent soil moisture condition (AMC) and precipitation (Hawkins, 1993). Besides this, the prediction of P-R relationship in ungauged regions was achieved utilizing the inter-gauged and ungauged basin parameter transfer method that was previously recommended in other hydrological studies as a reliable approach for parameter predictions in ungauged basins (PUB) (Bárdossy, 2007; Blöschl, 2006). Using this method, the gridded observed runoff coefficients (Orc) were transferred to ungauged regions according to their hydrologic similarity. Monthly hydrologic similarity's feature datasets were established from key runoff controlling factors such as: (i) AMC, (ii) NRCS-CN, (iii) terrestrial water storage change (TWSC), (iv) land-surface temperature (T), and (v) topographic parameters (topographic wetness index (TWI) and slope). The present study established a unique monthly hydrologic similarity feature dataset with multiple zones. Each zone is composed of a set of grids with similar climatic and physical characteristics. The runoff controlling factors runoff controlling factors were first classified into ranges, converted to non-simplified polygons and stacked together using an overlay (intersect) analysis technique (Zhu, 2016) performed with the "intersect tool" available in "overlay tools" which is one of the "Analysis tools" in ArcMap v.10.5. After that, the mean observed runoff coefficients were transferred to ungauged regions employing the "Zonal Statistics as Table Tool" available in "Zonal tool" of the "Spatial Analyst Tools" in ArcMap v.10.5" where, the zonal feature datasets of hydrologic similarity were considered as "Input raster or feature zone data", and gridded observed runoff coefficient as "Input value raster". Inter-gauged and ungauged basin parameter transfer approach were chosen to be used in this study because of its simplicity and reasonable prediction in ungauged regions, yielding the results representing a real-world phenomenon occurring in the same region. This method can be considered as one of the hybrid interpolation or gaps filling techniques which are very useful in developing various datasets such as temperature, precipitation, soil, etc.

Review comment 2

Below are additional general comments on the manuscript, also the paper does not use common hydrological terminology, which I have highlighted in the last section with detailed comments. General comments: Page 2, Line 10 "Although runoff studies have been conducted at global scale and in some local areas in Africa" Although there are few studies describing continental scale hydrology in Africa (but they do exist, see Schuol et al 2008), it is pertinent untrue to say there are only some studies on runoff in local areas in Africa. The authors are advised to perform a detailed literature review before writing the introduction and providing such an untrue statement.

Response

We would like to apologize for this inconsistent statement. In the revised manuscript a lot thing was changed and modified based on your comment. The runoff-related studies conducted on African case studies were quoted: "Various runoff-related studies have been carried out with different purposes such as, for example, runoff depth estimation at global scale (Hong et al., 2007;Fekete et al., 2002a) (Hong et al., 2007;Fekete et al., 2002b;Ruess, 2015;Smakhtin, 2004) and water stress assessment at country and global scales (Ruess, 2015;Smakhtin, 2004), modelling blue and green water availability in Africa (Schuol et al., 2008) and runoff predictions in different parts of Africa (Tesemma et al., 2010;Olang and Fürst, 2011;Jaleta et al., 2017;Mahmoud, 2014;Karamage et al., 2017a)."

Review comment 3

Page 2, Line 24-26 "The Rc data were then interpolated to the ungauged areas using the key factors such as land-surface temperature (T), Precipitation (P) and potential runoff coefficient (Co) estimated from the land use and land cover, soil texture, and slope information by using GIS spatial analysis techniques." This is a typical approach to regionalize hydrological processes in ungauged basins. The authors should refer to earlier studies under the PUB initiative of the IAHS to explain how this approach is a well-accepted approach. However, in this approach, I don't understand how the runoff

coefficient (Rc) is dependent on the runoff coefficient (Co)? Are these not the same thing?

Response

Thank you for your concern about the PUB initiative of the IAHS. Based on your comment the interpolation method was revised and more details were provided. Actually, the methodology used to predict the runoff depths and coefficients in ungauged regions can be considered as a hybrid interpolation method or inter-gauged and ungauged basin parameter transfer method based on spatial hydrological similarities analyzed using the key runoff controlling factors. As you already mentioned, it is one of the recommended approaches for hydrological predictions in ungauged basins (PUB) (Bárdossy, 2007; Blöschl, 2006; Chiew et al., 2018).

Review comment 4

Page 5 figure 2, the conceptual framework calculates the runoff coefficient from the rainfall and runoff databases, afterward it recalculates the runoff by multiplying the rainfall with the runoff coefficient (top right box), how is this relevant?

Response

We would like to apologize for the above-mentioned misstatement. The final results of the present study were generated at the long-term monthly and annual mean temporal resolution. The monthly runoff coefficients were estimated for each month whenever runoff discharge was recorded. Then, all historical monthly coefficients were summed and divided by the number of recorded months to obtain the long-term monthly average runoff coefficient for each station.

Review comment 5

Page 6, figure 3, colors indicated in the legend are not consistent with the map, unclear what the legend "mean number of streamflow recorded months per each month of the year during 1901-2017" means. Based on the text, this would mean 12 maps? Also,

I see some areas with yellow markers indicating less than 10 years of data, correct? And for the Congo river, I only see a marker at the downstream end of the basin. How does this affect the analyses? Also using data from any period seriously affects the analyses, as many dams have been constructed in the later decades. Even for stations with data across the 117 years, this needs to be taken into consideration.

Response

We are sorry for the unclear legend of figure 3. This figure was remapped and the legend was corrected. In the revised manuscript, runoff discharges for very large catchments were replaced by the sub-catchments with the medium size. Regarding the temporal gaps, 75.64% of the total observed extent comprise the runoff discharges with a record of more than 20 years (Figure 3). Pre-analysis of the historical changes in annual runoff discharges suggested a linear trend varies between 10% and 40% among the stations which drain the catchments covered by a small extent (8.44% total gauged area). Indeed, a large proportion (91.56%) of the total African gauged area, including the catchments recorded in earlier to recent decades has stations which experienced a minor variance ranging from 0% to 10% which is not a major problem in long-term bases analysis of runoff estimation. In our revised manuscript, monthly and spatial of water storage change within different parts of the continent were considered in hydrologic similarity analysis (Figure 7) using the terrestrial water storage changes estimated from the Center for Space Research (CSR) Gravity Recovery and Climate Experiment (GRACE) RL05 mascon solutions available at 1o resolution for the period starting from April 2002 to June 2016 (Save, Bettadpur, & Tapley, 2016). Except, the precipitation datasets available for since the beginning of 20th century, even before, the other above-mentioned changing runoff controllers are available for the recent decades (i.e.: GRACE data for water storage change analysis were collected since 2002 and good quality land cover maps are available since the 1990s). Lack of these data for the earlier decades constrained us to predict the past runoff process. Again, if the earlier runoff discharges are excluded from the long-term runoff calculations, spatial gaps

would be increased and bring more challenge for validation.

Review comment 6

Page 6 equation 1 is an obvious equation and does not need to be presented.

Response

Thank you for your concern about unnecessary equation presented in our previous manuscript. In the revised manuscript the indicated equation was removed.

Review comment 7

Page 7 line 1-10: using two different datasets may bring in additional uncertainty, is it really worth including the additional 3 years of data? Page 7 equation 2, does this mean you have for each station 12 Rc values for each month? The units for runoff and precipitation (line 14) should be mm/month. How do you convert this to an Rc on an annual basis?

Response

Thank you for your advice. In the revised manuscript, the data were reprocessed using an update GPCC rainfall products spanning a period between 1901 – 2016. The units for runoff and precipitation was also added as mm·month-1. The annual runoff depth (mm·yr-1) is the total of monthly runoff depths for all 12 months of a year. The average annual runoff coefficient is estimated as the ratio of annual runoff depth (mm·yr-1) to the annual precipitation intensity (mm·yr-1).

Review comment 8

How do you take into account different availability of data for specific months? Section 2.2.1: how many stations were used for the study with a typical availability of data?

Response

Thank you for these questions, monthly runoff coefficient was estimated for each month

that has runoff discharge. Then, all historical monthly coefficients were summed and divided by the number of recorded months to obtain a long-term monthly average runoff coefficient for each station. We have included antecedent soil moisture condition (AMC) estimated with recorded day precipitation for the period of more than 30 years. Accumulative AMC has the potential to separate the areas that have experienced different past climatic conditions. Also, the land use maps for the period of 25 years were utilized to isolate areas with different historical land cover changes. The available GRACE dataset that has been collected since 2002 were utilized to isolates regions with different terrestrial water storage changes due to climate and seasonal variability and anthropogenic activities (i.e.: water storage for hydropower generation and its release, irrigation, water consumption, etc).

Review comment 9

Section 2.2.2: why is there a completely different approach for ungauged basins? How are the two studies linked?

Response

Ungauged regions covered 52.57% of the total continent of Africa that seems to be larger extent compared to the recorded catchments (47.43% of African continent) (Figure 3) due to 31% of the continent occupied by the desert of Sahara (Cook & Vizy, 2015); the remaining ungauged regions account only 21.57% of the total African continent. The is a significant connection between the estimates over gauged and ungauged regions because the observed catchments are the source of data for ungauged regions. The inter-gauged and ungauged basin parameter transfer method can be considered as a hybrid interpolation method under the guidance of the spatial hydrologic similarity condition.

Review comment 10

Page 11&12 figure 6: how was the most right map developed from the three other maps

(not explained in the text) What would be interesting is to assess how the approach in 2.2.2 is able to generate Rc for the gauged basins to validate the approach used.

Response

Thank you for this comment. Hydrologic similarity conditions were investigated using the runoff controlling factors selected based on their potential impact highlighted in previous studies. Thus, the efficiency analysis of the approach used to predict the data for filling the gaps suggested that the estimated and observed runoff coefficients have the goodness of fit ($R^2$) ranging from 0.56 to 0.67 for the long-term monthly Rc and 0.78 for the annual mean Rc (Figure 14). These results are within permissible validity limits since an $R^2 > 0.5$ is considered acceptable for calibration and validation in hydrological modeling (Santhi et al., 2001; Van Liew et al., 2003).

Review comment 11

Page 13 line 11 it is very confusing when the authors use ETc in a different way it is normally used (for crop evapotranspiration)

Response

We are sorry for this unclear presentation of the equation. In the revised manuscript, the equation of ETc was express in words.

Review comment 12

Page 13,14,15 section 3: there is absolutely no reflection on what figure 6&7 are showing, is this the result of the methodology described in section 2.2.1 or 2.2.2?

Response

Thank you for highlighting the insufficient description on figure 6 and 7. In the revised manuscript detailed information was provided (section 2.2.2.3 and 3.1).

Comment 13

Also, observed runoff presented in figure 9 does not include the entire continent, how can this be compared to the interpolated one which covers the entire continent?

Response

Thank you for the concern about the unrealistic-data comparison in figure 9. This mistake was corrected in the revised manuscript.

Review comment 14

The observed basins often collect data from large river basins, which have different dominating processes compared to smaller basins.

Response

Thank you for this comment. We appreciate this valuable comment. In the revised manuscript, runoff discharges for very large catchments were replaced by the sub-catchments with the medium size. Also, observed runoff coefficients for 535 catchments covering about 47.43% of the whole continent were downscaled at 0.5° grid scale based on grids' direct runoff contributions to their corresponding basins estimated following the Natural Resources Conservation Service (NRCS) runoff curve number (CN) approach. NRCS-CN involves the precipitation, the land use and land cover (LULC) information, soil hydrological characteristics, and antecedent soil moisture condition (AMC) assessed according to antecedent precipitation index (API).

Review comment 15

Page 19, section 3.3 To estimate runoff coefficients on the country scale is irrelevant, as they do not constitute a drainage basin. Schuol, J., Abbaspour, K.C., Yang, H., Srinivasan, R. and Zehnder, A.J., 2008. Modeling blue and green water availability in Africa. Water Resources Research, 44(7).

Response

Thank you for raising this point. In the revised manuscript, the relevancy of runoff estimation on the non-catchment scale was highlighted. usually, runoff-related studies are often conducted at a drainage basin scale, but, hydrological studies on the grid and country scales are very useful at the national level since each government has own policies for water resource management. For instance, it has been noticed that runoff discharges are useful in water stress analysis on a country scale (Ruess, 2015; Smakhtin, 2004). Integration of NRCS-CN in downscaling the runoff discharges do not alter the quantity of observed runoff at a catchment scale, but it redistributes catchment's discharged runoff volume to its grids proportionally according to their respective climate and physical conditions

Review comment 16

Detailed comments: Use of definitions: Page 1 Line 28 "lacking precipitation" Line 30 "precipitation scarcity" Line 31 "eastern and western drylands of Africa" Page 2 Line 4 "water runoffs" Line 4 "hazards and disasters" Line 5-6 "flood threats" Line 12 "indicative runoff coefficient" what do you mean with indicative? (Chen, Liu, Li, & Wang, 2007; Sriwongsitanon & Taesombat, 2011).

Response

We apologize for these mistakes. In the revised manuscript, the above-mentioned mistakes were corrected.

Review comment 17

Line 17 "water flow" Line 19 "runoff coefficient is very useful for catchment scale land use and flood management", I disagree with this statement, floods are often associated with short timespans.

Response

We are sorry for this misstatement. In the revised manuscript, the importance of runoff coefficient was described as follows: "runoff coefficient is very useful for rainfall-runoff management in different land cover types since it can easily identify the ratio of rainwater flowed from each land use type under heterogeneous climate and physical conditions among different grids of the catchment. It may help to locate areas with high potential runoff risk which require special practices of stormwater management (Chen et al., 2007)."

Review comment 18

Line 30-31 "Evapotranspiration is generally less than precipitation in wet seasons, that is positive water balance due to groundwater accumulation, which results in increased surface runoff." I do not understand this sentence.

Response

We are sorry for this misstatement. In the revised manuscript, the introduction part was revised and the above-mentioned statement was removed.

Review comment 19

Line 32 "plants absorb underground water" why only the plants? Line 33-34 "underground water can be ignored in the long-term annual mean water balance" Do you mean that "the change in storage" can be ignored?

Review comment 20

We are sorry for the unclear statement. In the revised manuscript, the statement was revised as follows: "Underground water storage change also plays a significant role in runoff generation process throughout the alteration of soil moisture condition. However, in the long-term annual mean basis of water balance analysis, the estimation of terrestrial water storage change provides approximately zero values due to a variety of wet and dry seasons (Long, Longuevergne, & Scanlon, 2014)."

Review comment 21

Page 3 Line 8 unit for mean precipitation should be mm/year Source of data for figure 1? Page 4 Line 12-13 "Drainage patterns are controlled by the distribution of basins

and swells, about 95% is drained through permanent or ephemeral rivers" what other types of rivers exist? And what is a swell? Aren't drainage patterns dependent on geographical location, topography, climatological factors etc? Line 14 "sand sea"??

Response

We are sorry for this incomplete statement. In the revised manuscript, the introduction part was revised and the above-mentioned statement was removed.

Again, we are thankful and appreciate your constructive comments.

Please also note the supplement to this comment:
https://www.hydrol-earth-syst-sci-discuss.net/hess-2018-424/hess-2018-424-AC3-supplement.zip